



# A parameterisation for the co-condensation of semi-volatile organics into multiple aerosol particle modes

Matthew Crooks[1], Paul Connolly[1], and Gordon McFiggans[1]

[1]The School of Earth, Atmospheric and Environmental Science, The University of Manchester, Oxford Road, Manchester, M13 9PL

*Correspondence to:* matthew.crooks@manchester.ac.uk

**Abstract.**

A new parameterisation for cloud droplet activation of multiple aerosol modes is presented that includes the effects of co-condensation of semi-volatile organic compounds (SVOCs). The novel work comes from the dynamic condensation parameterisation that approximates the partitioning of the

SVOCs into the condensed phase at cloud base. The dynamic condensation parameterisation differs from equilibrium absorptive partitioning theory by calculating time dependent condensed masses that depend on the updraft velocity. Additionally, more mass is placed on smaller particles than at equilibrium, which is in better agreement with parcel model simulations. All of the SVOCs with saturation concentrations below $1 \times 10^{-3} \ \mu g^{-3}$ are assumed to partition into the condensed phase

at cloud base, defined as 100% relative humidity, and the dynamic condensation parameterisation is used to distribute this mass between the different aerosol modes. An existing cloud droplet activation scheme is then applied to the aerosol particles at cloud base with modified size distributions and chemical composition to account for the additional mass of the SVOCs. Parcel model simulations have been performed to test the parameterisation with a range of aerosol size distributions, composi-

tion and updrafts. The results show excellent agreement between the parameterisation and the parcel model and the inclusion of the SVOCs does not degrade the performance of the underlying cloud droplet activation scheme.

## 1 Introduction

Clouds make an important contribution to both weather and climate and so understanding the com-

plex physical processes involved in their formation and continued existence is crucial for long term weather and climate modelling. The size and number concentration of cloud droplets can significantly alter a cloud's albedo by changing the amount of reflected shortwave radiation and absorbed longwave radiation (Twomey, 1991; McCormick and Ludwig, 1967; Chýlek and Coakley Jr, 1974).



Cloud lifetime is also tightly coupled to albedo (Twomey, 1974, 1977) as well as directly to cloud
droplet properties, through precipitation rate (Stevens and Feingold, 2009).

One of the most significant factors that influences cloud droplet number is the properties of aerosol
particles from which cloud droplets are formed by condensation of water under supersaturated conditions (Pruppacher and Klett, 1977). An increase in number concentration of such cloud condensation
nuclei (CCN) can lead to an increase in cloud droplet number due to the higher abundance of par-
ticles for water to condense onto. Conversely, in some situations, larger particles, which typically
activate at lower relative humidities, can deplete available water and inhibit the activation of smaller
particles within the population (Twomey, 1959; Ghan et al., 1998). With such differing consequences
resulting from variations in aerosol particle properties it is no surprise that the largest cause of uncertainty in global mean radiative forcing is attributed to aerosol-cloud interactions (Lohmann et al.,
2000); contributing an estimated -0.4 Wm$^{-2}$ to -1.8 Wm$^{-2}$ (Carslaw et al., 2013; Forster et al.,
2007).

Accurate representation of the cloud droplet activation process is, therefore, of crucial importance.
Global weather and climate models, however, are not only restricted by our understanding of the microphysical processes involved but, additionally, by the computational expense required to model
them. Several cloud activation parameterisations have been developed to predict cloud droplet number as a function of aerosol properties (Ming et al., 2005; Abdul-Razzak et al., 1998; Abdul-Razzak
and Ghan, 2000; Shipway and Abel, 2010; Fountoukis and Nenes, 2005; Nenes and Seinfeld, 2003).
Despite an emphasis on computational efficiency these parameterisations have been largely successful (Ghan et al., 2011; Simpson et al., 2014). The most popular are those by Abdul-Razzak and Ghan
(Abdul-Razzak et al., 1998) and Fountoukis and Nenes (Fountoukis and Nenes, 2005). The former
was originally only tested up to a mean radius of 0.1 $\mu$m, which is where it shows some deviation
from parcel model simulations. Both have been shown to perform well up to 250 nm at high number
concentrations with a tendency to overpredict at lower number concentrations (Simpson et al., 2014).
The Fountoukis and Nenes parameterisation was later extended to account for kinetic limitations of
larger droplets (Barahona et al., 2010). Under a wide parameter space the Fountoukis and Nenes
parameterisation, with the giant CCN (Barahona et al., 2010) extension at larger particle sizes, is
found to perform better than the Abdul-Razzak and Ghan (Simpson et al., 2014; Connolly et al.,
2014) and, consequently, this is the only activation parameterisation studied in this paper.

The parameterisations mentioned above assume that the aerosol particles are entirely involatile.
Although it is common to make such an assumption for primary emissions, regional and global scale
studies of semi-volatile primary organic aerosol have been carried out with mostly improved estimates of the organic aerosol budget (Tsimpidi et al., 2010, 2014; Pye and Seinfeld, 2010; Jathar
et al., 2011). Secondary organic aerosol (SOA) is formed by nucleation of new particles from organic
vapours or by condensation of oxidation products of precursor gases into the particle phase. Subse-
quent particle-phase reactions age the condensed compounds to produce compounds that are less





volatile or functionally involatile. The condensation of semi-volatile organic compounds (SVOCs) onto aerosol particles increases their size, changes their chemical composition and consequently affects their ability to act as CCN. Depending on the geographical location, between 5% and 90% of the total aerosol mass can be composed of organic material (Andreae and Crutzen, 1997; Zhang

et al., 2007; Gray et al., 1986) with a significant, but uncertain, proportion made up of SOA. Any realistic cloud activation scheme must, therefore, include the effects of SVOCs in the formation of SOA.

Direct chemical and dynamical modelling of every organic species is not only computationally impractical due to the many thousands of different organic species present in the atmosphere (Gold-

stein and Galbally, 2007) but is rendered impossible by only a small fraction of these having been identified (Simpson et al., 2011; Borbon et al., 2013). To facilitate numerical modelling, large numbers of compounds are commonly grouped together and are represented by fewer surrogate species with effective chemical properties (O'Donnell et al., 2011).

Equilibrium absorptive partitioning theory was introduced by Pankow (1994) to calculate the equi-

librium vapour/condensed phases of volatile compounds and is often used in global models as a computationally efficient approximation to the dynamically evolving vapour/condensed phases. The empirically derived relation of Odum et al. (1996) was introduced as a method of treating multiple organic species based on the results of two-compound experiments and while it benefits from its simplicity it has been found to be unrealistically sensitive to changes in the concentration of the

organic compounds (Cappa and Jimenez, 2010). The volatility basis set of Donahue et al. (2006) allows many organic species to be binned according to their saturation concentration. This was later extended to include the condensation of water (Barley et al., 2009) and also introduces a new molar definition of the saturation concentration. A recent advance (Crooks et al., 2016) has extended this molar based approach to calculating the equilibrium condensed concentrations across multiple

aerosol modes of different sizes and chemical composition when each particle contains a non-volatile constituent. In some applications, this can also be applied to particles that have previously nucleated from extremely low volatility compounds by approximating the resulting aerosol mass as involatile (Ehn et al., 2014).

The work of Connolly et al. (2014) proposes the only extension of the aforementioned cloud acti-

vation schemes to include the effects of SVOCs. The parameterisation assumes that the vapour and condensed phases of the SVOCs are in equilibrium at cloud base, which is a reasonable approximation in all but very low number concentrations of aerosol particles and high updrafts. Equilibrium at cloud base is calculated using equilibrium absorptive partitioning together with a $\log_{10}$ volatility basis set at a relative humidity of 99.999%. The condensed phase is assumed to have resulted from

the condensation onto a single mode of non-volatile particles with sizes distributed according to a lognormal. The additional mass has the effect of changing the median diameter and geometric standard deviation in such a way as to preserve mass while keeping the arithmetic standard deviation





constant. The new particle size distribution and composition at cloud base is then inserted into the
Abdul-Razzak et al. (1998) and Fountoukis and Nenes (2005) parameterisations in order to calculate
the number of cloud droplets.

Although the new parameterisation of Connolly et al. (2014) was found to agree well with a parcel
model it does have the limitation of only applying to a single aerosol particle mode, which is too
restrictive for most atmospheric situations. In this paper, we extend the parameterisation of Connolly
et al. (2014) to the case of multiple aerosol modes, which is significantly more complicated. While
the SVOCs may be in bulk equilibrium at cloud base, as in the single mode case, the time scale re-
quired for the condensed masses on each mode to reach equilibrium is on the order of several hours.
In many situations, the aerosol particles may activate before the condensed phase of the SVOCs has
equilibrated between the different sizes of particles. The result is that the multiple mode equilib-
rium partitioning of Crooks et al. (2016) can miscalculate the condensed masses achieved under the
dynamic conditions experienced in cloud droplet activation, especially at high updraft speeds. We
present a new parameterisation to calculate the condensed masses of SVOCs across multiple aerosol
modes during the rapid dynamic condensation induced by high relative humidities near cloud base.
These condensed masses at cloud base are then used in the cloud activation scheme (Fountoukis and
Nenes, 2005) in an analogous way to the equilibrium condensed masses in the single mode case
(Connolly et al., 2014). At low updrafts there is more time for the condensed masses to equilibrate
before activation resulting in the new dynamic parameterisation producing the same results as using
the multiple mode equilibrium theory. The new parameterisation is found to significantly outperform
the equilibrium model at higher updrafts when compared to a dynamic parcel model.

## 2   Dynamic condensation parameterisation description

We describe here the model that is applied to approximate the partitioning of the condensed masses
of SVOCs at cloud base. For brevity we use the acronym DCP for this dynamic condensation pa-
rameterisation. The condensation of the SVOCs is based on the principle that if the vapour pressure
exceeds the equilibrium vapour pressure then there is net condensation and if it deceeds equilibrium
then the organic compound undergoes net evaporation. During long-range aerosol transport this can
lead to the vapour and condensed phases equilibrating as shown by the lower three inset boxes in
Figure 1. In the case of cloud droplet activation, the rapid condensation of water can suppress the
equilibrium partial pressure of the organics to a negligible level, which is demonstrated in Section
2.1. This simplifies the condensation rate of the SVOCs to be proportional to their partial pressure.
In addition, it causes the organics to undergo continuous condensation until their vapour phase is
depleted. The left three inset boxes in Figure 1 show this process. As the aerosol particle rises in the
atmosphere, water and the organics condense onto the particle, reducing the mass of SVOCs in the
vapour phase and, consequently, reducing the condensation rate. In cloud, the condensed mass of




water increases drastically and causes all of the organics to condense into the particle phase. In the
following section, we justify this assumption through an example before deriving the parameterisa-
tion for both monodisperse and polydisperse aerosol populations.

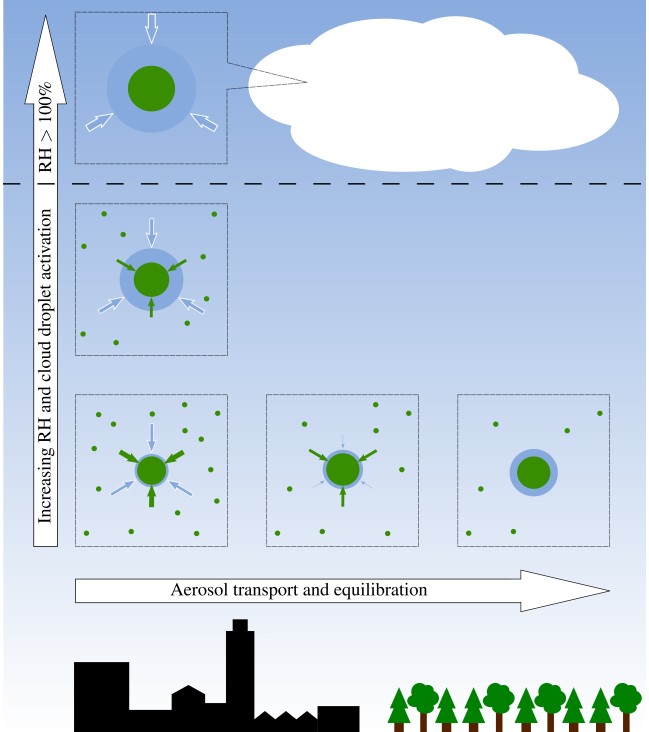

**Figure 1.** Schematic of the two different condensation processes occurring in the atmosphere with condensation
rates of organics and water indicated by the size of the green and blue arrows, respectively. The lower three inset
boxes show the equilibration between the vapour and condensed phases of the semi-volatile organic compounds
during large-scale aerosol transport. Gas phase species changes due to the air mass passing over anthropogenic
and biogenic emission sources, as well as oxidation and gas phase reactions. The left three boxes show the rapid
depletion of the vapour phase of organics that results from the relative humidity reaching and exceeding 100%.
Ground based emissions sources of SVOCs do not contribute additional vapour mass to the rising air parcel
represented by the left 3 inset boxes.

## 2.1   Neglecting the equilibrium partial pressure

Ideality is assumed in order to calculate the equilibrium partial pressure of both water and the or-
ganics. Consequently, the equilibrium saturation ratio of both water and the organic compounds can
be described by the mole fraction of the condensing compounds multiplied by a Kelvin term. The
condensed mass of water in the mole fraction is calculated assuming water condenses sufficiently





quickly to be perpetually in equilibrium. At the high relative humidities experienced near cloud base the denominator of the mole fraction is dominated by the condensed water and reduces it to negligible values in activated particles compared to the actual saturation ratio.

To demonstrate the suppression in equilibrium partial pressure of the organics we have run a parcel model with binned microphysics (Topping et al., 2013) that solves the dynamic condensation numerically. For this example, we have used an aerosol population composed of three lognormal size distributions given by the natural environmental conditions discussed in Section 5. We used an SVOC mass loading of 27 $\mu$g m$^{-3}$ and the vertical wind speed is 2 m s$^{-1}$.

Figure 2 shows the ratio of the equilibrium partial pressure of the three median diameters to the saturation partial pressure as a function of relative humidity. On the second and third modes, which activate into cloud drops, this ratio drops to below 0.2 and 0.1, respectively, as the relative humidity approaches 100%. The first mode, however, has a much larger value and actually increases above 1 near 100% RH. This is a result of this mode not activating and so does not have a significant quantity of water condensed on it to suppress the equilibrium partial pressure of the organics.

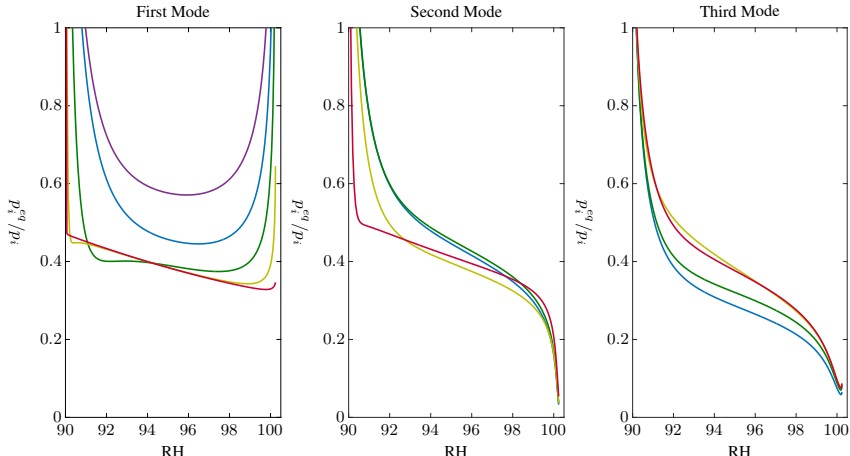

**Figure 2.** The ratio of the equilibrium partial pressure, $p_i^{eq}$, to the partial pressure, $p_i$, of each of the organic compounds on each of the modes as a function of relative humidity. Each colour shows a different volatility bin. On the first mode the compounds with the lowest five $C^*$ values look indistinguishable from each other and are all shown by the purple line. Similarly the six compounds with the lowest $C^*$ values in the second and third plots are shown by the blue line.

## 2.2 A monodisperse aerosol particle population

We first assume that the aerosol particle population is composed of $N$ particles of diameter $D_d$. The condensation rate of the $i^{th}$ organic compound onto a particle is assumed to be proportional to the difference between the partial pressure, $p_i$, and the equilibrium partial pressure, $p_i^{eq}$, of the $i^{th}$





compound over the particle,

$$\frac{dy_i}{dt} = \alpha_i D \left(p_i - p_i^{eq}\right). \tag{1}$$


Here $y_i$ is the condensed mass per unit mass of air and $t$ is time. The variable $\alpha_i$ is defined as

$$\alpha_i = \frac{2\pi D_{v,i} M_i}{RT},$$

where $M_i$ and $D_{v,i}$ are the molecular weight and diffusivity of the organic compound in air, respectively. The universal gas constant is denoted $R$ and $T$ is the temperature. $D$ is the wet diameter of the particle, which is calculated assuming the condensed water is in equilibrium at the initial RH. For simplicity we set the temperature and pressure to their initial values in the parameterisation, which is common in cloud droplet activation parameterisations (Fountoukis and Nenes, 2005; Abdul-Razzak et al., 1998). Therefore, in the parameterisation the variable $\alpha_i$ becomes a constant parameter. The diameter, $D$, however, varies with time as the SVOCs condense onto and evaporate off the particle. This variation is highly non-linear and for simplicity we assume that for the parameterisation that setting $D$ to the initial wet diameter is sufficiently accurate an approximation.



The saturation ratio of the $i^{th}$ organic compound can be expressed as the ratio of the mixing ratio, $r_i$, to the saturation mixing ratio, $r_i^{sat}$, or equivalently, the ratio of the partial pressure to the saturation partial pressure, $p_i^{sat}$. Equating these two definitions gives

$$\frac{p_i}{p_i^{sat}} = \frac{r_i}{r_i^{sat}}. \tag{2}$$


For simplicity, we denote the ratio of the saturation partial pressure to the saturation mixing ratio of the $i^{th}$ organic compound by $\beta_i$ so that (2) simplifies to

$$p_i = \beta_i r_i. \tag{3}$$

The initial mixing ratio, $r_i^0$, corresponding to a condensed mass of $y_i^0$, can be related to the initial partial pressure, $p_i^0$, through equation (3); namely $p_i^0 = \beta_i r_i^0$. The mixing ratio subsequently decreases at the same rate as the increase in total condensed mass concentration of each compound. Hence


$$r_i = r_i^0 - N(y_i - y_i^0).$$

Substituting this into equation (3) yields

$$p_i = p_i^0 - N\beta_i(y_i - y_i^0), \tag{4}$$


Substituting equation (4) into (1) and neglecting the equilibrium partial pressure produces

$$\frac{dy_i}{dt} = \alpha_i D \left(p_i^0 + N\beta_i y_i^0 - N\beta_i y_i\right). \tag{5}$$

This can be integrated assuming constant diameter, temperature and pressure to give

$$y_i = y_i^0 + \frac{p_i^0}{N\beta_i} \left(1 - e^{-\alpha_i D N \beta_i t}\right). \tag{6}$$





Equation (5) represents an analytic approximation to the time evolution of the condensed mass of

the SVOCs on a single particle within a monodisperse aerosol. The evolution of the total condensed

mass can be obtained by multiplying $y_i$ by $N$. Thus

$$Y_i = Ny_i^0 + \frac{p_i^0}{\beta_i}\left(1 - e^{-\alpha_i DN\beta_i t}\right). \tag{7}$$

All terms on the right-hand side of equations (6) and (7), except $t$, are parameters that depend on the

initial conditions of the problem, such as the temperature, pressure, and initial condensed mass of

the organic compounds. The only time dependence is in the exponential term.

### 2.3   Multiple monodisperse aerosol particle populations

In order to be applicable in atmospherically relevant situations, the approximation in the previous

section needs to be extended to polydisperse aerosols. Suppose now that the particle population is

composed of multiple monodisperse populations of diameters $D_{d,j}$ and number concentrations $N_j$.

In this case, we have an equation analogous to (1) for each size of particle,

$$\frac{dy_{ij}}{dt} = \alpha_i D_j\left(p_i - p_{ij}^{eq}\right). \tag{8}$$

We note that both the partial pressure and the parameter $\alpha_i$ are independent of size but the equi-

librium partial pressure is dependent on $D_j$, although, like in the monodisperse case, we neglect

this.

The change in partial pressure resulting from condensation of the SVOCs is proportional to the

total condensed mass across all particles. Hence, the evolution of the partial pressure, analogous to

(4), and is given by

$$p_i = p_i^0 - \beta_i \sum_k N_k\left(y_{ik} - y_{ik}^0\right), \tag{9}$$

where $k$ is a dummy index used for the summation over $j$ to distinguish from the equations for the

$j^{th}$ size of particle. The initial condensed mass of the $i^{th}$ organic compound on a particle of size

$D_j$ is denoted $y_{ij}^0$. Substituting equation (9) into (8) and neglecting the equilibrium partial pressure

results in the equation

$$\frac{dy_{ij}}{dt} = \alpha_i D_j\left(p_i^0 - \beta_i \sum_k N_k\left(y_{ik} - y_{ij}^0\right)\right). \tag{10}$$

We now multiply by $N_j$ and sum over $j$, again using the dummy index $k$

$$\frac{d}{dt}\left(\sum_k N_k y_{ik}\right) = \alpha_i \gamma\left(p_i^0 - \beta_i \sum_k N_k\left(y_{ik} - y_{ij}^0\right)\right). \tag{11}$$

For simplicity, we have denoted

$$\gamma = \sum_k N_k D_k.$$





The total condensed mass of the $i^{th}$ compound across all particles is denoted $f_i$ and can be expressed as

$$f_i = \sum_k N_k y_{ik},$$

and, similarly, the initial total condensed mass is given by

$$f_i^0 = \sum_k N_k y_{ij}^0.$$

Equation (11) can now be simplified as

$$\frac{df_i}{dt} = \alpha_i \gamma \left( p_i^0 - \beta_i f_i + \beta_i f_i^0 \right).$$

This equation is qualitatively similar to equation (5) in the monodisperse case, (5), and so the solution can be expressed in an analogous way as

$$f_i = f_i^0 + \frac{p_i^0}{\beta_i} \left( 1 - e^{-\alpha_i \gamma \beta_i t} \right). \tag{12}$$

To calculate the condensed mass on each of the individual particles we substitute (12) into equation (10) to give

$$\frac{dy_{ij}}{dt} = \alpha_i D_j p_i^0 e^{-\alpha_i \gamma \beta_i t}, \tag{13}$$

which can be integrated directly to produce

$$y_{ij} = y_{ij}^0 + \frac{D_j p_i^0}{\gamma \beta_i} \left( 1 - e^{-\alpha_i \gamma \beta_i t} \right). \tag{14}$$

Equation (14) expresses the time evolution of the condensed mass of each compound in a particle within a population composed of multiple monodisperse modes. To obtain the total condensed mass on a particular monodisperse mode, $Y_{ij}$, this expression can be multiplied by $N_j$,

$$Y_{ij} = N_j y_{ij}^0 + \frac{N_j D_j p_i^0}{\gamma \beta_i} \left( 1 - e^{-\alpha_i \gamma \beta_i t} \right). \tag{15}$$

### 2.4 Polydisperse aerosol particle populations

Typically, atmospheric aerosol particles occur in a continuous range of sizes. The continuous size distribution may be discretised into collections of similarly sized particles to create an aerosol particle population that is composed of multiple monodisperse modes and which approximates the continuous size distribution. Alternatively, in many situations the continuous distribution of particle sizes can be represented by one or more lognormal size distributions as defined by the equation

$$\frac{dN}{d\ln D} = \sum_j \frac{N_j}{\sqrt{2\pi} \ln \sigma_j} \exp \left[ - \left( \frac{\ln \left( \frac{D}{D_{m,j}} \right)}{\sqrt{2} \ln \sigma_j} \right)^2 \right]. \tag{16}$$

Equation (16) denotes the number concentration of particles per natural logarithm of the bin width. Here, $N_j$ is the total number of particles represented by the $j^{th}$ lognormal and $\ln \sigma_j$ and $D_{m,j}$ are





the geometric standard deviation and median diameter, respectively. The advantage of representing a polydisperse particle population in this way is that each lognormal size distribution can be treated as a single mode by replacing the diameters, $D_j$, in equation (14) by the median diameters and multiplying by the total number of particles in each mode. Hence the condensed mass on a lognormal mode is approximated by the expression

$$Y_{ij} = N_j y_{ij}^0 + \frac{N_j D_{m,j} p_i^0}{\gamma \beta_i} \left( 1 - e^{-\alpha_i \gamma \beta_i t} \right). \tag{17}$$

### 2.5 Fractional representation

As the condensed mass of an organic compound increases in the full time-dependent equation, (8), the difference between the partial pressure and the equilibrium partial pressure decreases and this slows the rate of condensation. Eventually, the condensation rate approaches zero as the condensed mass approaches the equilibrium value. In the parameterisation, the equilibrium partial pressure has been neglected and this process does not happen. The parameterisation, however, has been derived to be applicable specifically when the relative humidity is close to 100% for the purpose of approximating cloud droplet activation, convergence on equilibrium in this case is not directly of relevance. An additional and more important problem arises in the high RH regime, however. In this case, the equilibrium vapour partial pressure of the organic compounds is close to zero; the partial pressure of an organic compound decreases as the condensed mass increases and eventually becomes zero when all of the compound has entered the condensed phase. Due to the approximations to the diameter, temperature and pressure, the partial pressure does not decrease at the correct rate and can reach zero with either too little mass in the condensed phase or even calculate a condensed mass that exceeds the total abundance of that compound. This violates conservation of mass. To maintain mass within the system, we introduce a fractional formulation that approximates what fraction of the condensed mass exists in each mode as a function of time. To do this, we divide the condensed masses given by (15) by the sum of $Y_{ij}$ over all particles, thus

$$Z_{ij} = \frac{N_j y_{ij}^0 + \dfrac{N_j D_j p_i^0}{\gamma \beta_i} \left( 1 - e^{-\alpha_i \gamma \beta_i t} \right)}{\sum\limits_k \left( N_k y_{ik}^0 + \dfrac{N_k D_k p_i^0}{\gamma \beta_i} \left( 1 - e^{-\alpha_i \gamma \beta_i t} \right) \right)}. \tag{18}$$

To calculate the distribution of the condensed mass at cloud base, the fractional formulation, (19), can be evaluated at cloud base and then multiplied by the total abundance of SVOCs.

In the lognormal mode case, the fractional formulation takes the form

$$Z_{ij} = \frac{N_j y_{ij}^0 + \dfrac{N_j D_{mj} p_i^0}{\gamma \beta_i} \left( 1 - e^{-\alpha_i \gamma \beta_i t} \right)}{\sum\limits_k \left( N_k y_{ik}^0 + \dfrac{N_k D_{mk} p_i^0}{\gamma \beta_i} \left( 1 - e^{-\alpha_i \gamma \beta_i t} \right) \right)}. \tag{19}$$





## 3 Cloud drop activation parameterisation

### 3.1 Single aerosol mode

The parameterisation employed to calculate the number of cloud droplets including the effects of SVOCs is a modification to that described in Connolly et al. (2014). In this earlier work the system was seeded with a single involatile mode whose particle sizes could be represented by one lognormal size distribution. A description of the methodology is given here before the theory is extended to the multiple mode case.

The initial temperature, pressure and relative humidity are prescribed; in this paper we use the values 293.15K, 95000Pa and 90%, respectively. All aerosol particles are assumed to contain an involatile constituent so that no particle can evaporate completely. It is assumed that the SVOC vapours and the involatile particles have coexisted for sufficient time for the condensed masses to be in equilibrium at 90% RH, calculated using a molar based equilibrium absorptive partitioning theory (Barley et al., 2009). The additional mass from the condensed SVOCs is added to the involatile mass and the new particle sizes are assumed to follow a lognormal size distribution with the same geometric standard deviation as the involatile particles but an increased median diameter that is calculated to conserve mass. Further details are given in Appendix A1 or, alternatively, the original paper (Connolly et al., 2014).

In the single mode case, the condensed masses of the semi-volatile organic compounds are assumed to be in equilibrium with the vapour phase at cloud base (99.999% RH) and are calculated using equilibrium absorptive partitioning theory. This additional aerosol mass is added to the initial composite aerosol and a new median diameter and geometric standard deviation of the dry aerosol size distribution are calculated which conserve mass and maintain a constant arithmetic standard deviation, as defined by

$$SD = e^{\ln D_m + \frac{1}{2} \ln \sigma} \sqrt{e^{\ln^2 \sigma} - 1}. \tag{20}$$

More details on deriving the size distribution at cloud base are given in Appendix A2. These new aerosol size distribution parameters are then input into a widely used parameterisation for cloud droplet activation in the absence of SVOCs (Fountoukis and Nenes, 2005), which is already constructed to accept multiple modes.

### 3.2 Multiple aerosol modes

In the multiple mode case we consider polydisperse aerosol particle populations that can be represented by multiple lognormal size distributions. As in the single mode case, we assume that the initial condensed masses of each SVOC are in equilibrium with the vapour phase and these are calculated using multiple mode equilibrium absorptive partitioning theory (Crooks et al. (2016)). The initial median diameter of each mode is calculated in the same way as in the single mode case; assuming



conservation of mass together with the same geometric standard deviation as the involatile particle

modes.

Bulk condensation of SVOCs into multiple lognormal modes is qualitatively similar to the single lognormal case and, as such, bulk equilibrium between a condensed and vapour phase at cloud base is still achieved. It can take several hours, however, for the condensation and evaporation of SVOCs between the different modes to reach equilibrium. Therefore, it cannot be assumed that the individual

condensed masses at cloud base are in equilibrium even though the assumption of bulk equilibrium still holds. Rather than calculating bulk equilibrium using multiple mode equilibrium partitioning theory and summing over each mode it is quicker, and not significantly less accurate, to simply assume that all of the SVOCs are in the condensed phase at cloud base. The parameterisation for dynamic condensation of SVOCs, described in the Section 2, can then be used to calculate how this

mass partitions between each aerosol particle mode.

An additional complication with using the DCP solution in the multiple mode case, rather than equilibrium partitioning theory, is that there is a time dependence and the solution changes depending on how long the particles experience elevated RH values near cloud base before activating. This is largely determined by the vertical updraft. The time, $t_{cb}$, that it takes for a parcel of air to reach 100%

RH from our initial value of 90% assuming a linear relative humidity, temperature and pressure profile can be calculated and is described in Appendix B. The difference in condensed mass of the SVOCs only changes significantly when the RH is close to 100%. In addition, the DCP solution was derived under the condition of constant temperature and pressure, as well as relative humidities close to that at cloud base. Consequently, the DCP solution needs to be evaluated at time that is shorter

than $t_{cb}$. We have found that evaluating the DCP solution at the time it takes for the RH to increase form 99.9% to 100% yields the best results at updrafts above about 1 m s$^{-1}$, although it is largely insensitive at slower updrafts. Initial condensed masses are still calculated using the initial RH. A review of this is presented in Appendix B and the supplement.

## 4    Results

**Table 1.** Total mass loadings of organics used in Section 4 that are distributed between the volatility bins using the anthropogenic volatility distribution given in Table 3. Concentrations are given in $\mu$g m$^{-3}$ at 293.15 K and 950 hPa.

| Figure | top-left | top-right | lower-left |
|--------|----------|-----------|------------|
| 3 | 0.913 | 2.09 | 4.83 |
| 4 | 1.92 | 4.43 | 10.4 |
| 5 | 2.35 | 5.45 | 12.8 |



The simulations are initiated at 95% RH, a temperature of 293.15K and a pressure of 95000 Pa. The condensed concentrations of the SVOCs are assumed to be in equilibrium with the vapour phase initially and the median diameter increased to conserve mass accordingly. The geometric standard deviation of the initial composite aerosol is the same as the ammonium sulphate modes.

    We neglect nucleation of new particles from the SVOC vapours as these are unlikely to grow large

enough to activate into cloud droplets. In addition, the rapid growth of existing particles during the cloud droplet activation process will induce significant condensation of the SVOCs that will act as the dominant sink of the organic vapours. We further assume in the parcel model simulations that the initial condensed SVOCs remain in the condensed phase throughout the cloud activation process. As the relative humidity increases monotonically up to the point of activation it is likely that

the further condensed water will act to increase the condensed mass of SVOCs across all particles with minimal evaporation. The only particles that are likely to undergo evaporation of SVOCs are the smallest particles whose condensed water is scavenged by the larger particles. These particles, however, have only a small amount of condensed SVOCs compared to the mass of the larger particles, especially near cloud base, and this additional mass will have a limited effect on the activation

of larger particles.

    Figures 3 to 5 show the fraction of the total number of aerosol particles that activate. The shape of the size distributions of ammonium sulphate are shown in the lower right plots and the total number concentration of particles is varied between the three associated plots. Parameter values used in each plot are given above each graph. To show the enhancement in cloud droplet number as a result of the

SVOCS, the fraction of activated drops both with and without condensing vapours are shown by the green and blue, respectively. The crosses show the parcel model results while the solid lines show the parameterisation. The results from two alternative methods of including the SVOCs are also compared. The first, shown by the dashed red line, applies multiple mode equilibrium absorptive partitioning theory (Crooks et al., 2016) at cloud base to distribute the SVOCs between the different

modes, rather than the DCP. This method is analogous to the original SVOC parameterisation of Connolly et al. (2014). The second method includes the initial condensed concentration of SVOCs, at the temperature and pressure from which the activation scheme of (Fountoukis and Nenes, 2005) is applied, but without additional co-condensation of the remaining vapours and is shown by the dashed grey line.

Figure 3 shows the result of having six times more aerosol particles in the smaller mode than the larger. At all three particle number concentrations, the inclusion of just the initial condensed mass of SVOCs (dashed grey) produces little enhancement in CCN concentration compared to the case when no organics are considered (blue line). The additional co-condensation of SVOC vapours that occurs when the RH is above 95% is clearly important in this case. The assumption that the condensed mass

of SVOCs is in equilibrium across all modes at cloud base results in more particles in the larger-sized mode activating at lower updrafts, below 0.1 m s$^{-1}$, but at higher updrafts the activated fraction is



similar to the case without co-condensation. The new DCP shows a pronounced enhancement across all updrafts, shown by the green line. Below 1 m s$^{-1}$ in the upper left plot, the DCP captures the enhancement in CCN concentration calculated in the parcel model well but at higher updrafts it over-predicts. It is worth noting, however, that some of this over-prediction is attributable to the activation scheme of (Fountoukis and Nenes, 2005), which also over-predicts the CCN concentration even in the absence of SVOCs. At the higher particle number concentration, shown in the top right plot, the DCP parameterisation is in excellent agreement with the parcel model and outperforms the other models significantly. All the parameterisations under-predict the CCN concentrations at very high particle number concentrations, shown by the lower left plot. The DCP parameterisation does out-perform the others and is the only one that predicts an enhancement by the SVOCs at higher updrafts, which is also seen in the parcel model.

The size distribution used in Figure 4 has a larger portion of the particle number concentration in the smaller mode than in Figure 3. In this case the DCP parameterisation performs better than in Figure 3 at lower number concentrations. The application of equilibrium absorptive partitioning theory at cloud base appears to agree with the parcel model at lower updrafts and predicts a signifi-cant enhancement in CCN concentration compared to the case without SVOCs. At higher updrafts, however, assuming equilibrium of the SVOCs at cloud base activates a similar number of particles to case with water only. Neglecting the co-condensation of the organic vapours during activation, again, produces little enhancement in CCN concentration. The new DCP performs very well at the lower number concentrations used in the upper two plots across all updrafts but, as was seen in Figure 3, none of the parameterisations predict a significant enhancement at high number concentrations. Even in the absence of SVOCs, the parameterisation of Fountoukis and Nenes (2005) under-predicts compared to the parcel model and may indicate an over-sensitivity of the underlying cloud droplet activation scheme to high number concentrations.

The size distribution used to generate Figure 5 has two modes containing equal numbers of par-ticles. The performance of all of the parameterisations is qualitatively similar to in Figure 4. The assumption of equilibrium of the SVOCs at cloud base predicts a large enhancement in CCN from the larger-sized mode but has little effect on the activation of the smaller particles. The DCP performs the best across all updrafts at lower number concentrations but at higher number concentrations the parcel model predicts a significant enhancement in number of CCN that is not captured by any of the parameterisations.

Figure 6 directly compares the activated fractions from the DCP parameterisation against the parcel model from Figures 3 to 5. The crosses show just the Fountoukis and Nenes (2005) parame-terisation without any SVOCs while the dots show the effect of the SVOCs using the DCP. The dots are coloured by total particle number concentration and it is clear that the parameterisation under-predicts compared to the parcel model at high number concentrations, as previously discussed. The agreement at lower number concentrations, however, is largely similar between the parameterisation





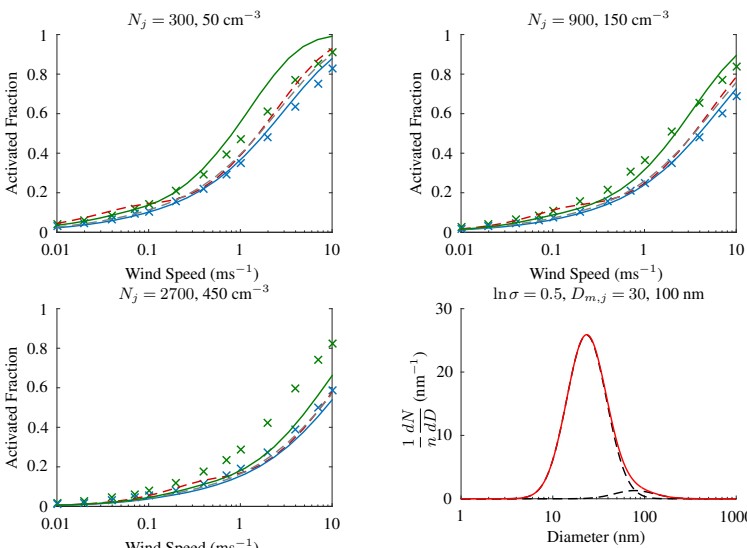

**Figure 3.** Fraction of total aerosol particles that activate into cloud droplets at a range of vertical updrafts and number concentrations. The solid lines show the results from the parameterisation and the crosses show the parcel model results while the effect of the SVOCs are shown by the green and the blue shows the analogous results without the organic vapours. The red dashed lines show the effect of applying equilibrium absorptive partitioning at cloud base and the dashed grey line shows the results of only including the initial condensed concentration of SVOCs. The shape of the size distribution of ammonium sulphate is shown in the lower right plots with median diameters and geometric standard deviations written above. Number concentrations used are specified above each plot.

with SVOCs and without, indicating that the new parameterisation does not degrade the performance
of the underlying cloud droplet activation scheme.

## 5   Environmental Variations

We now test the parameterisation for different environmental conditions by varying the aerosol size distributions and volatility distribution. Different aerosol size distributions are considered with parameter values taken from Van Dingenen et al. (2004), a study that has gathered together field data
from multiple sites across Europe to obtain typical measurements for four different environmental conditions; Natural, Rural, Near-city and Urban. Each environment varies in proximity to major anthropogenic sources of pollution by distances of >50 km, 10-50 km, 3-10 km for the first three, respectively. The Urban sites are defined as having fewer than 2500 vehicles per day within a 50 m radius.





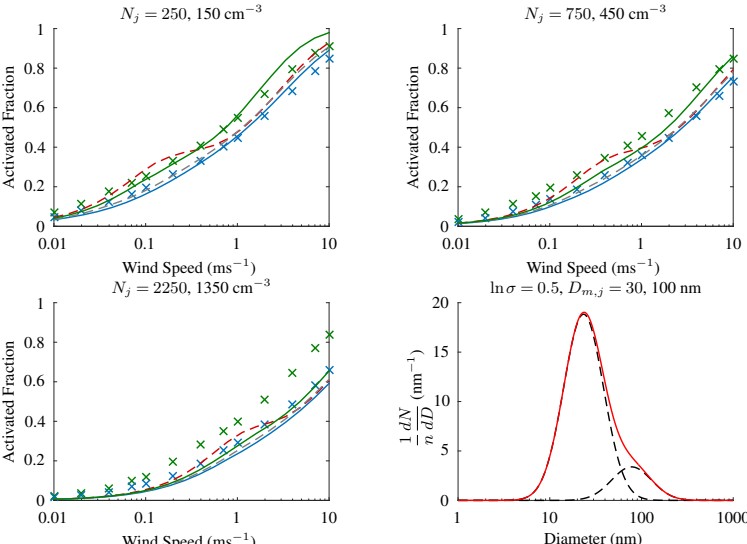

**Figure 4.** Same as Figure 3 but with a different size distribution

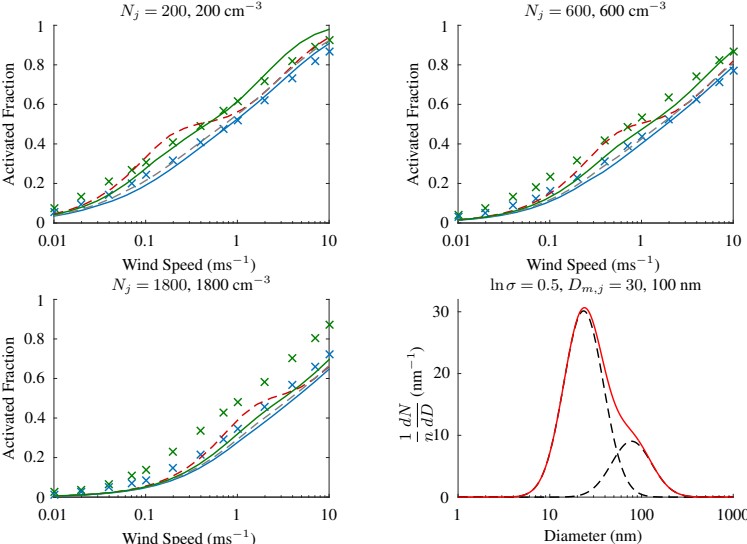

**Figure 5.** Same as Figure 3 but with a different size distribution

Table 2 shows the summertime afternoon number concentration, $N$, median diameter, $D_m$, and geometric standard deviation, $\ln \sigma$, from Van Dingenen et al. (2004) obtained by fitting three lognormal size distributions to the data. It is clear from the values of $N_1$ that there are significantly higher





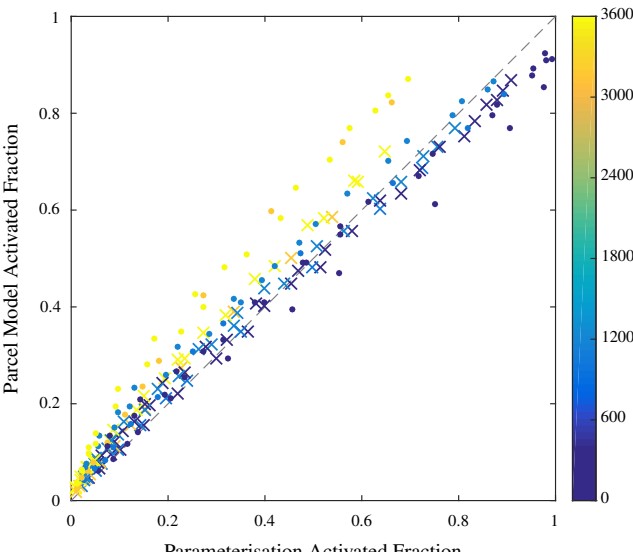

**Figure 6.** Activated fraction data from Figures 3 to 5 collated into one figure. The $x$ axis shows the activated fraction from the parameterisation and the $y$ axis shows the parcel model. The crosses show the results from the cloud droplet activation scheme without SVOCs and the dots show the parameterisation with SVOCs. Markers are coloured by total number concentration of aerosol particles.

concentrations of very small particles in the Near-city and Urban environments than the Natural and Rural. This indicates that these small particles are a result of the anthropogenic sources, most

likely combustion engines. Particles of less than 50 nm are typically created as a result of particle nucleation during the initial cooling phase of vehicle emissions (Kittelson (1998), Harris and Maricq (2001), Myung and Park (2012)) with 90% of the number concentration being in this nucleation mode (Kittelson (1998)). This is exemplified in the figures in table 2 with 85 % and 95% of the Near-City and Urban number concentrations, respectively, being in the first two modes with median

diameters less than 50 nm.

**Table 2.** Number concentration (# cm$^{-3}$), median diameter (nm) and geometric standard deviation for the four cases studied in Van Dingenen et al. (2004) during summertime afternoons.

| Case | Mode 1 | | | Mode 2 | | | Mode 3 | | |
|---|---|---|---|---|---|---|---|---|---|
| | $N_1$ | $D_{m,1}$ | $\ln \sigma_1$ | $N_2$ | $D_{m,2}$ | $\ln \sigma_2$ | $N_3$ | $D_{m,3}$ | $\ln \sigma_3$ |
| Natural | 185 | 26 | 0.44 | 1364 | 85 | 0.47 | 276 | 246 | 0.32 |
| Rural | 2089 | 28 | 0.45 | 553 | 60 | 0.30 | 1459 | 114 | 0.52 |
| Near-City | 2938 | 13 | 0.66 | 3989 | 32 | 0.69 | 1356 | 123 | 0.54 |
| Urban | 7751 | 11 | 0.57 | 5422 | 41 | 0.69 | 599 | 189 | 0.43 |



Putaud et al. (2004) is an accompanying paper to Van Dingenen et al. (2004) that further analyses
the aerosol properties in the different environments to provide aerosol composition. We assume that
the smaller two modes are composed of a carbonaceous aerosol particles. As much as 80% of emitted
black carbon is hydrophobic (Cooke et al. (1999)) and will not contribute towards cloud droplet
activation. While in the atmosphere, however, these particles undergo an ageing process involving
oxidation, coating with sulphate and SOA and photochemical decomposition (Rokjin et al. (2005),
Zuberi et al. (2005), Zuberi et al. (2005)), which results in a hydrophilic composition. The degree
to which these particles have aged and their resulting chemical composition varies greatly with time
and location and, as such, the possible values that can be assigned to the material properties of these
particles has a large variability.

In the current study of cloud droplet activation, assuming newly formed, insoluble, hydrophilic,
pure black carbon particles will not contribute to the CCN number concentration and therefore will
not provide an interesting analysis. Instead, we attempt to choose parameter values that represent
aged hydrophilic particles. Molecular weights of several compounds found in newly-formed par-
ticulate matter from combustion range from 178 - 302 g mol$^{-1}$ (Miguel et al. (1998) Peaden et al.
(1980), Allen et al. (1996)). We choose a value of 200 g mol$^{-1}$. The density of atmospheric carbona-
ceous aerosols have been found to lie in the range 1 - 1.7 g cm$^{-3}$ (Spencer et al. (2007), Slowik et al.
(2004), Svenningsson et al. (2006)) and a value of 1.5 g cm$^{-3}$ is used here. To simulate hydrophilic,
nondissociative particles we choose a van't Hoff factor of 1; this is in line with values found for
levoglucosan (Zarra et al. (2009), Svenningsson et al. (2006)). The larger mode in all four regions
is modelled as composed of ammonium sulphate and this is chosen to represent all highly soluble
compounds in a single mode of particles that will act as effective CCN.

We use two different volatility distributions across the four sites; one representing biogenic sources
and the other anthropogenic sources. The former will be used for the Natural and Rural sites and the
latter for the Near-City and Urban sites. The biogenic volatility distribution is taken from the 1DVBS
of Hermansson et al. (2014) which uses the model of Simpson et al. (2012) to distribute oxidation
products of $\alpha$-pinene across nine volatility bins with $\log \mathcal{C}^*$ values ranging from $10^{-5}$ to $10^3$ $\mu$g
m$^{-3}$, separated by factors of 10. We assume for our modelling study that the molar based $\mathcal{C}^*$ can
be obtained from the mass based $\mathcal{C}^*$ by dividing by the molecular weight of the compounds in the
volatility bin. The volatility distribution is similar across all three sites in Hermansson et al. (2014),
with little more than a rescaling in total concentration between them, and so, without loss of general-
ity, we take the values from Abisko in Northern Sweden; these values are given in Table 3. Biogenic
SOA can mostly be composed of compounds with molecular weights in the region of 130 g mol$^{-1}$
(Gao et al. (2004), Henze and Seinfeld (2006)) although much higher molecular weight compounds
have been identified (Kourtchev et al. (2015)). A density of 1.4 g cm$^3$ is used and is taken from Gao
et al. (2004).




The anthropogenic volatility distribution is taken from Cappa and Jimenez (2010) which is derived from field measurements in Mexico City. Again, the volatility bins are separated by orders of magnitude in $\log \mathcal{C}^*$ but range from $10^{-6}$ to $10^3$ $\mu$g m$^{-3}$. We choose typical values in the literature for hydrocarbons produced in vehicular combustion engines (Miguel et al. (1998) Peaden et al. (1980), Allen et al. (1996)). A density of 1.25 g cm$^{-3}$ and molecular weight of 200 g mol$^{-1}$ are used, together with a van't Hoff factor of 1.

**Table 3.** Volatility distributions used to represent typical biogenic and anthropogenic SVOC concentrations. These values are rescaled in order to obtain the required organic mass fraction in the simulations. Concentrations are given in $\times 10^{-2}$ $\mu$g m$^{-3}$.

| Case | $\log \mathcal{C}^*$ | | | | | | | | | |
|---|---|---|---|---|---|---|---|---|---|---|
| | $10^{-6}$ | $10^{-5}$ | $10^{-4}$ | $10^{-3}$ | $10^{-2}$ | $10^{-1}$ | $10^0$ | $10^1$ | $10^2$ | $10^3$ |
| Biogenic | - | 0.1 | 0.1 | 0.5 | 0.5 | 4 | 10 | 20 | 44 | 32.5 |
| Anthropogenic | 0.5 | 1 | 2 | 3 | 6 | 8 | 16 | 30 | 42 | 80 |

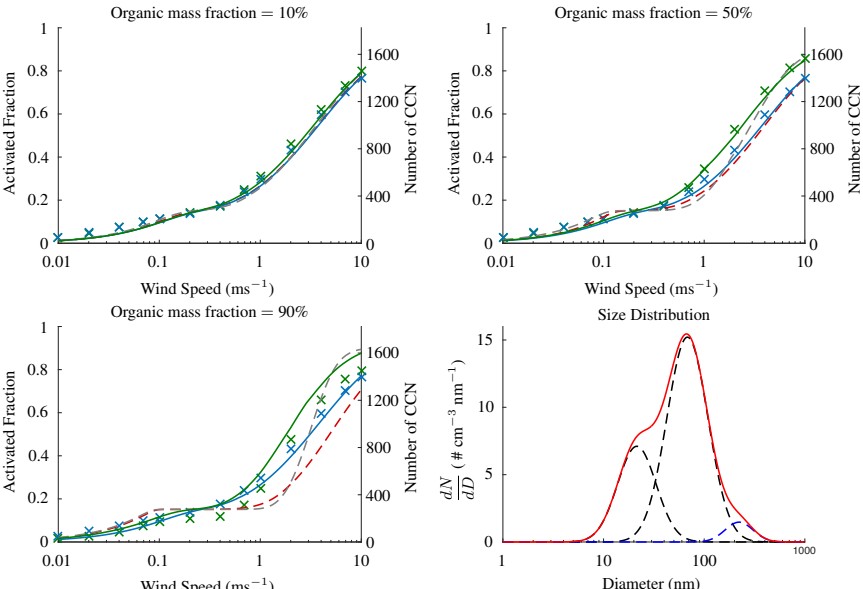

**Figure 7.** Activated fraction of aerosol particles using the Natural environment particle size distribution at a range of vertical updrafts. Three different concentrations of SVOCs are used and is specified above each plot. The crosses show the results from the parcel model and the solid lines show the parameterisation both with (green) and without SVOCS (blue). The effect of assuming multiple mode equilibrium at cloud base instead of the DCP are shown in dashed red and the dashed grey shows the activated fraction neglecting co-condensation of SVOCS during the ascent up to cloud base.



Three concentrations of SVOCs are investigated and each is obtained by rescaling the volatility distributions given in Table 3. Multiple mode equilibrium partitioning is used to calculate the condensed masses as 90% RH and the bulk organic mass fraction of the aerosol particles excluding water is calculated. The volatility distribution is then rescaled until the organic mass fraction is equal to 10%, 50% and 90%. Values of 10% and 50% are chosen in line with upper and lower limits frequently encountered. Organic mass fractions as high as 90% have been measured (Andreae and Crutzen, 1997; Zhang et al., 2007; Gray et al., 1986) but not all of this will be attributed to condensed SVOCs. Our simulation using 90% organic mass fraction of SVOCs is therefore used as an extreme, but still realistic, scenario to see how the parameterisation performs under a wide parameter space. Actual volatility distributions are given in appendix C and we note that the 50% organic mass fractions give total SVOC mass loadings of about 27 $\mu$g m$^{-3}$ for the Natural and Rural sites and 37 $\mu$g m$^{-3}$ for the Near-City and Urban environments. These are in line with the total mass loadings measured in Cappa and Jimenez (2010).

For comparison, we model each case with both the parcel model and the parameterisation. The parcel model is initiated with equilibrium condensed masses in the particle phase and new lognormal size distributions are calculated assuming the same geometric standard deviation as given in Table 2 for particle modes with median diameters above 50 nm. For smaller modes, we calculate a new geometric standard deviation that maintains a constant arithmetic standard deviation. This has been found to be more realistic under atmospheric time scales (Crooks et al. (2016)) although it does not seem to have a significant effect on cloud droplet number. Initial condensed masses are assumed to be involatile and the vapour phase is free to condense with increasing altitude and condensed water.

Two further parameterisations are presented for comparison. The first assumes the initial condensed mass of SVOCs is in equilibrium but does not include any additional condensation in the ascent to cloud base. This model is used to demonstrate the importance of co-condensation of SVOCs near cloud base on cloud droplet number in addition to the effect of SOA that may be measured at lower RH values. In order to justify the use of the new DCP we additionally show the results from the parameterisation assuming multiple mode equilibrium (Crooks et al. (2016)) at cloud base, which would be a direct analogy to the single mode parameterisation (Connolly et al. (2014)). This demonstrates the dynamic nature of the condensation process near cloud base.

Figure 7 compares the activated fraction of aerosol particles between the parameterisations and the parcel model simulations for the Natural environment. The new parameterisation with the DCP is shown by the solid lines with the green showing the effect of SVOCs and the blue showing the analogous results without SVOCs. The analogous results from the parcel model results are shown by the corresponding coloured crosses. As might be expected, low concentrations of SVOCs have a limited effect on cloud droplet number and this is demonstrated by the similarity between the green and blue solutions in the upper left plot. This is true for both the parcel model and the parameterisation, which are both also in good agreement for all wind speeds.



With concentrations of SVOCs corresponding to an organic mass fraction of 50% there is a pronounced increase in cloud droplet number as a result of the SVOCs at wind speeds above about 0.5 m s$^{-1}$. Above 1 m s$^{-1}$ vertical updraft, an additional 10% of the total number of particles activate. This corresponds to as much as a 20% relative increase in cloud droplet number, or 200 cm$^{-3}$, as a result of the SVOCs. Across the full updraft range there is excellent agreement between the parcel

model and the parameterisation. If the Fountoukis and Nenes (2005) parameterisation is used together with multiple mode equilibrium absorptive partitioning at cloud base (red dashed), however, there is a suppression in cloud droplet number compared to the cases when there are no SVOCs. Similarly, if only the condensed SVOCs at 90% RH are taken into account (grey dashed) there is also a small suppression between 0.2 and 2 m s $^{-1}$ indicating that in this range the co-condensation

of SVOCs near cloud base is most important and creates a narrower size distribution of particle sizes, which leads to an increase in cloud droplet number.

    In the lower left plot, corresponding to very high concentrations of SVOCs, the parameterisation begins to deviate slightly from the parcel mode and does not pick up the suppression in cloud droplet number between 0.2 and 1 m s $^{-1}$. The agreement between the parcel model and the parameterisation

is, overall, still good considering the extreme abundance of SVOCs. The discrepancies in the additional two parameterisations shown by the grey and red dashed lines seen in the upper right plot are further demonstrated here but to a much more severe level. In the cloud base equilibrium case, there is a significant suppression in cloud droplet number of magnitude similar to the enhancement due to the SVOCs seen in the DCP parameterisation and the parcel model simulation. The co-condensation

of SVOCs near cloud base are even more important at higher concentrations and a significant suppression is seen around 1 m s$^{-1}$ vertical wind speed. This suppression occurs at much higher vertical updrafts and to a much higher extent than is seen in the parcel model.

    Comparisons of cloud droplet number for the Rural environment case between the parcel model and the parameterisation are shown in Figure 8. There are, overall, a smaller fraction of the particles

activating than in the Natural environment. Due to there being twice as many particles in the Rural case, however, this translates into roughly the same number concentration of CCN. Furthermore, at 10 m s$^{-1}$ there are only 10% of particles that do not activate in the Natural case and this corresponds to the smallest mode. The smallest mode in the Rural case has roughly the same size distribution as in the Natural case but contains half of the total number of particles. These, again, do not activate to

produce the maximum activated fraction of 0.5 at high updrafts

    The enhancement in number of CCN reaches a maximum of about 300 cm$^{-3}$ in the 50% organic mass fraction case at higher updrafts. This is a similar relative increase of 20% compared to the case without SVOCs that was seen in the Natural environment that had an analogous increase of 200 cm$^{-3}$ and slightly lower number of CCN. The agreement between the parcel model and the

parameterisation is excellent with a very slight underestimate at lower updrafts. The enhancement at higher updrafts is captured almost exactly. At very high concentrations, however, the parcel model





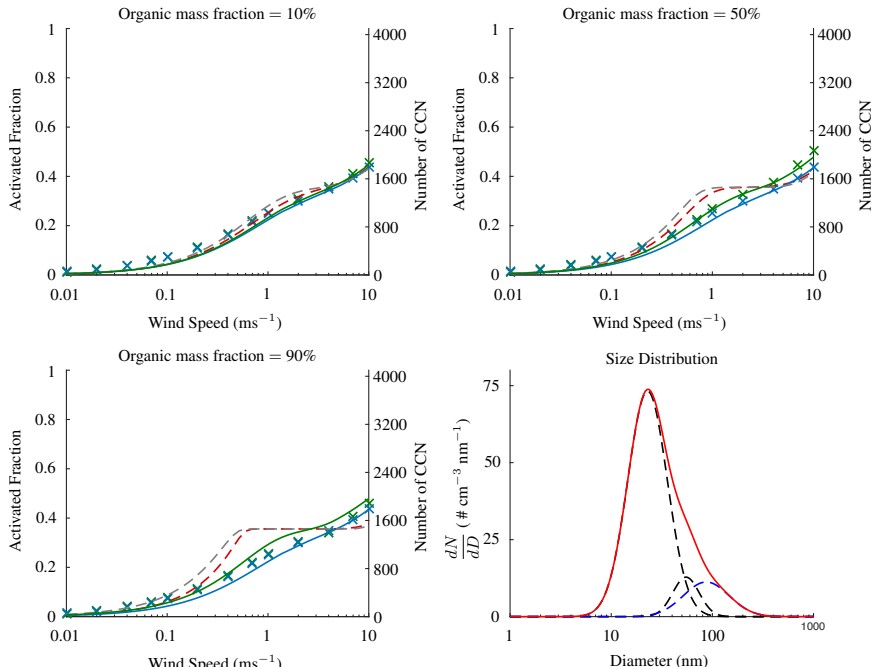

**Figure 8.** Same as Figure 7 for the Rural environment.

predicts very little increase in number of CCN from the SVOCs while the parameterisation shows a noticeable enhancement. Overall, the agreement is still very good and the errors from the new parameterisation with SVOCs are not larger than those resulting from the Fountoukis and Nenes

parameterisation without SVOCs.

The importance of the DCP is, again, very prevalent in Figure 8. The cloud base equilibrium parameterisation (dashed red) significantly overpredicts the number of CCN at lower updrafts due to there being more condensed mass on larger particles at equilibrium than is seen under dynamic conditions. This increases the size of the largest particles too much allowing them to activate at lower

supersaturations. The same effect can be seen in Figure 7 below 0.2 m s$^{-1}$. Conversely, there is less condensed mass of organics on smaller particles and this increases the supersaturation required for activation. Therefore, at higher updrafts, when smaller particles begin to activate in the parcel model, the cloud base equilibrium parameterisation underpredicts the number of CCN. Neglecting the co-condensation of SVOCs near cloud base produces similar activated fractions to the cloud base

equilibrium parameterisation. This is due to a significant additional condensed mass on smaller particles occurring from the co-condensation of SVOCs that creates a larger sink of water on unactivated particles and suppresses the supersaturation.




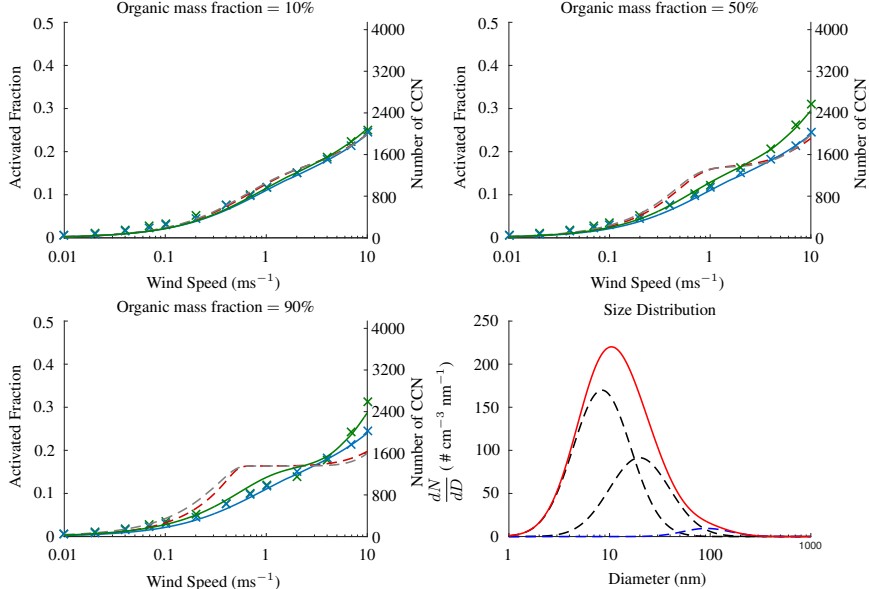

**Figure 9.** Same as Figure 7 for the Near-City environment.

Figure 9 shows the results from the Near-City case and look very similar to the Rural environment. In this case, however, more particles activate at higher updrafts but this corresponds to a lower

activated fraction. At very high updrafts there is pronounced increase in number of CCN caused by the first and second modes beginning to activate. For a vertical updraft of 10 m s$^{-1}$, this produces an enhancement in cloud droplet number of as much as 500 cm$^{-3}$ compared to the case without SVOCs. Again, the parameterisation without the DCP activate the first mode at much lower vertical updrafts than the parcel model.

In the Urban environment, shown in Figure 10 the SVOCs have little effect at low concentrations, as was seen in the three other cases. The SVOCs have little effect on cloud droplet number below an updraft of 1 m s$^{-1}$ and this is likely to be the result of the fact that only the largest mode activates here, and this mode contains a very small proportion of the total number of aerosol particles. Above 1 m s$^{-1}$, however, there is a drastic increase in number of CCN calculated in the parcel model and

the parameterisation with the DCP. The parcel model and the parameterisation are both in excellent agreement with both calculating an enhancement of 1000 cm$^{-3}$ in the upper right plot. In the lower left plot, the parameterisation predicts an enhancement of 1000 cm$^{-3}$ which is lower than the 1500 cm$^{-3}$ from the parcel model. Overall, the agreement is still very good. Above 1 m s$^{-1}$, the additional two parameterisations without the DCP predict a suppression in cloud droplet number resulting from

the SVOCs. As was previously discussed, this is a result of DCP partitioning a significant amount of the condensed mass at cloud base onto the smaller particles, which facilitates activation.



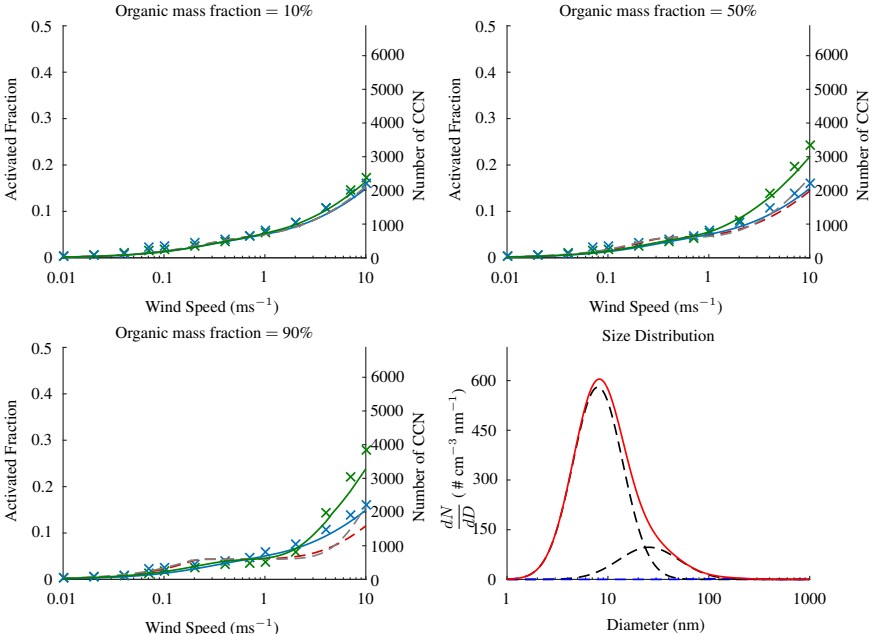

**Figure 10.** Same as Figure 7 for the Urban environment.

All of the activated fraction data from Figures 7 to 10 are collated in Figure 11. The $x$ axis shows the data from the parcel model and the $y$ axis shows the parameterisation with the DCP. As might be expected from Figures 7 to 10, the agreement in activated fraction is very good with all data points

lying close to the 1:1 line. The black dots, corresponding to the Urban environment, show a little deviation around an activated fraction of 0.2 but these data points correspond to the highest updrafts and the highest concentration of SVOCs, which are both extreme cases.

## 6  Conclusions

A parameterisation of the cloud droplet activation process including the effects of SVOCs is pre-

sented. The novel dynamic condensation parameterisation (DCP) provides an analytic approxima-
tion to the co-condensation of SVOCs near cloud base. In particular, it describes how the condensed mass of SVOCs is distributed between different aerosol modes. This is crucial for cloud droplet ac-
tivation as it is important to predict the change in particle sizes and chemical composition in order to ascertain the critical supersaturation required for activation.

In the paper, we have presented results using equilibrium absorptive partitioning theory to dis-
tribute the condensed SVOCs across the different modes and, in general, this approach places too much mass on larger modes than is observed in the detailed parcel model. Consequently, the larger





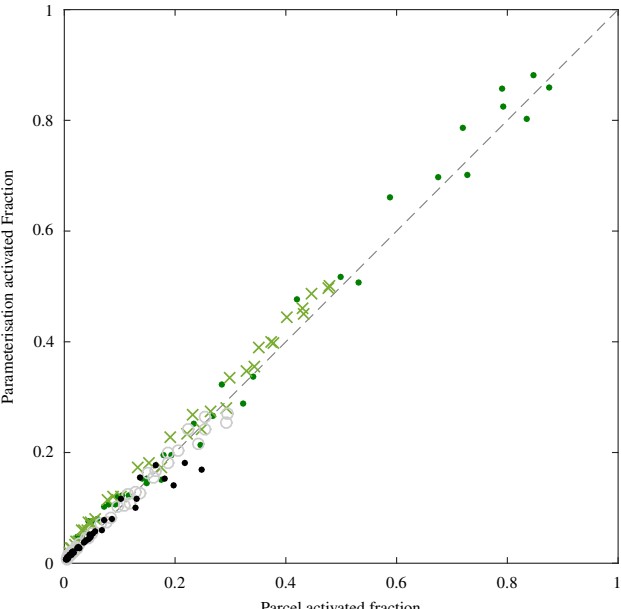

**Figure 11.** Scatter plot comparing the activated fractions from the parcel model against the parameterisation for the four different environments; Natural (green dots), Rural (green crosses), Near-City (grey circles) and Urban (black dots). The parcel model data is the same as the green crosses in Figures 7 to 10 and the parameterisation data has been interpolated onto the same wind speeds as the parcel model. The 1:1 line is shown by the grey dashed line.

particles can activate at lower supersaturations, as well as the smaller particles requiring higher supersaturations to activate. This can result in a suppression in concentration of CCN compared to the case without SVOCs in simulations when the parcel model predicts an enhancement. The new DCP offers much better agreement with the parcel model in these cases.

Four different environmental aerosol populations have been investigated: Natural, Rural, Near-City and Urban. In general, the aerosol particles become smaller in size and higher in concentration in the more urbanised regions and this reduces the fraction of particles that act as CCN. Up to 80% - 90% of particles activate in the Natural environment compared to only 20% - 30% in the Urban. The number of CCN in these two cases differs by only a factor of two, however. In general, the inclusion of SVOCs produces an enhancement in concentration of CCN in all environments with the only exception being a suppression when the vertical updraft is around 0.5 m s$^{-1}$ in the very high concentration of SVOC cases.

The most prominent enhancement in CCN concentrations is seen at higher updrafts, resulting in a 20% increase of 200 cm$^{-3}$ in the Natural environment and a 50% increase of 1000 cm$^{-3}$ in the Urban environment. Significant underestimation in CCN concentrations of 50% at higher updrafts can result





from either neglecting the co-condensation of SVOCs during cloud droplet activation or assuming the condensed phase of the organics reaches equilibrium. Ignoring the dynamic condensation of
SVOCs at cloud base at lower updrafts can produce a gross overestimation in CCN concentrations by a factor of 2.

## 7 Code Availability

Matlab versions of the dynamic condensation parameterisation and its associated cloud droplet activation scheme are available at doi:10.5281/zenodo.801398.

## Appendix A: Change in size distribution parameters

In the parameterisation, we assume non-volatile seed particles are lognormally distributed. The additional aerosol mass from the condensed SVOCs makes all particles larger, but different sizes of particles increase by differing amounts. We assume that after the condensation of the SVOCs the particle sizes are still lognormally distributed and describe, in this section, how the median diameter
and geometric standard deviation can be calculated from the non-volatile particle size distribution parameters and the mass of condensed organics. Depending on how close to equilibrium the condensed masses of the SVOCs are between the different sizes of particles, the standard deviation of the lognormal can vary. When close to equilibrium, the size distribution is relatively wide with the larger particles increasing in size by more than the small particles. This regime is described in Ap-
pendix A1 and is used to calculate the equilibrium size distribution used at the start of the parcel model simulations and the paramaterisation. Under shorter time scales, the dynamic condensation process of the SVOCs increase the diameter of the smaller particles by more than the larger particles to produce a narrower size distribution. This scenario is assumed at cloud base and is described in Appendix A2.

## A1    Initial size distribution

The median diameter and geometric standard deviation of the involatile seed particles are denoted $D_m$ and $\ln \sigma_m$, respectively. For a total number concentration of $N$, the volume of the involatile constituent is

$$V_m = \frac{N\pi}{6} e^{3\ln D_m + \frac{9}{2}\ln^2 \sigma_m}. \tag{A1}$$

The total volume of the composite aerosol at the initial 90% relative humidity is obtained by adding the total condensed mass of the SVOCs to $V_m$. We denote the initial mass of the organics in the $i^{th}$ volatility bin by $m_i^0$, calculated using equilibrium absorptive partitioning theory. The total volume





of the aerosol can then be calculated using

$$V_T = V_m + \sum_i \frac{m_i^0}{\rho_i},$$
(A2)

where $\rho_i$ is the density of the SVOCs in the $i^{th}$ volatility bin. To calculate the initial median diameter of the composite aerosol, $D_0$, we assume a lognormal size distribution with a geometric standard deviation of $\ln \sigma_m$ and eliminate $V_m$ from (A2) using (A1) to give

$$\frac{N\pi}{6} e^{3\ln D_0 + \frac{9}{2}\ln^2 \sigma_m} = \frac{N\pi}{6} e^{3\ln D_m + \frac{9}{2}\ln^2 \sigma_m} + \sum_k \frac{m_i^0}{\rho_i}.$$
(A3)

This equation can then be solved to find $D_0$, which together with the geometric standard deviation

of $\ln \sigma_m$, defines the initial size distribution of the particles including the condensed SVOCs.

**A2    Cloud base size distribution**

We denote the new median diameter and geometric standard deviation of the dry aerosol size distribution at cloud base as $D_{cb}$ and $\ln \sigma_{cb}$, respectively. The expression for conservation of mass at cloud base, analogous to equation (A3), is

$$\frac{N\pi}{6} e^{3\ln D_{cb} + \frac{9}{2}\ln^2 \sigma_{cb}} = \frac{N\pi}{6} e^{3\ln D_m + \frac{9}{2}\ln^2 \sigma_m} + \sum_k \frac{m_i}{\rho_i}.$$
(A4)

Here, $m_i$ is the condensed mass of the the organic compounds in the $i^{th}$ volatility bin at cloud base, calculated using the DCP. Both $D_{cb}$ and $\ln \sigma_{cb}$ are unknowns and an additional equation is required in order to calculate their values. The arithmetic standard deviation, given by (20) is evaluated at 90% RH and is the equated to the analogous quantity at cloud base (100% RH) to give

$$e^{\ln D_0 + \frac{1}{2}\ln^2 \sigma_m} \sqrt{e^{\ln^2 \sigma_m} - 1} = e^{\ln D_{cb} + \frac{1}{2}\ln^2 \sigma_{cb}} \sqrt{e^{\ln^2 \sigma_{cb}} - 1}.$$
(A5)

Together, equations (A4) and (A5) form a set of simultaneous equations for $D_{cb}$ and $\ln \sigma_{cb}$ and can now be solved to find the median diameter and geometric standard deviation of the dry aerosol size distribution at cloud base.

**Appendix B:  Cloud base time**

In order to calculate the time at which cloud base is reached and, consequently, the time at which $y_{ij}$ should be evaluated, we assume a linear relative humidity profile with gradient given by its initial value. We further assume linear temperature and pressure profiles.

The water mixing ratio is defined as

$$r = \frac{\epsilon e}{P - e},$$
(B1)

where $e$ is the water partial pressure and $\epsilon = \frac{R_a}{R_v}$ is the ratio of the gas constants of dry air and water vapour, respectively. Assuming hydrostatic balance gives the pressure gradient

$$\frac{dP}{dt} = -\frac{P}{R_a T} gw.$$
(B2)



Here, $g$ is acceleration due to gravity and $w$ is the updraft velocity. If we evaluate this expression at the initial temperature, $T_0$, and pressure, $P_0$, then the linear pressure profile can be written as

$$P = P_0 - \frac{P_0}{R_a T_0} gwt, \tag{B3}$$


assuming constant updraft velocity.

Under subsaturated conditions there is little condensation of water and so the water mixing ratio remains approximately constant. As a result the release of latent heat from the water condensing is negligible compared to the decrease in temperature caused by a decreasing pressure with altitude.

The rate of change of temperature within a parcel of air can therefore be expressed as

$$\frac{dT}{dt} = \frac{R_a T}{P c_p} \frac{dP}{dt},$$

where $c_p$ is the specific heat of air at a constant pressure. Substituting in the pressure profile from (B2) gives

$$\frac{dT}{dt} = -\frac{gw}{c_p}.$$

This can be integrated to give a linear temperature profile of

$$T = T_0 - \frac{gwt}{c_p}. \tag{B4}$$

The initial water mixing ratio can be calculated from the definition of saturation ratio, S,

$$S = \frac{r}{r_{sat}},$$

where $r_{sat}$ is the water mixing ratio at saturation and can be calculated from (B1) using the saturation

vapour pressure. The initial water mixing ratio is then defined as

$$r_0 = \frac{S_0 \epsilon e_{sat}(T_0)}{P_0 - e_{sat}(T_0)},$$

where $S_0$ is the initial saturation ratio and $e_{sat}$ is the saturation vapour pressure, which can be calculated using, for example, the Clausius Clapeyron equation. To calculate a linear saturation ratio profile with time we calculate $S$ at some later time, $\delta t$, using the linear temperature and pressure

profiles of equations (B4) and (B3). First define the temperature and pressure at time $\delta t$ by

$$
\begin{aligned}
T_1 &= T_0 - \frac{gw\delta t}{c_p}., \\
P_1 &= P_0 - \frac{P_0}{R_a T_0} gw\delta t.
\end{aligned}
$$

The water mixing ratio at time $\delta t$ can then be calculated from (B1)

$$r_1 = \frac{\epsilon e(T_1)}{P_1 - e(T_1)}.$$

Hence the saturation ratio as a function of time can be approximated by

$$S = S_0 + \frac{t(r_1 - r_0)}{r_{sat}\delta t}. \tag{B5}$$





By defining cloud base as 100% RH, the time to reach cloud base can then be found by setting $S = 1$ in (B5) and rearranging to find $t$, thus

$$t_{cb} = \frac{(1 - S_0) r_{sat} \delta t}{r_1 - r_0}.$$

Here, $t_{cb}$ is the approximate time that it takes for a parcel of air to reach cloud base from the initial relative humidity. The DCP is derived to apply only in the case when the relative humidity is close to 100%, while cloud droplet activation parameterisations are often initiated at 90% RH. This leads to a disparity in the time domain when the two parameterisations are applicable. In the cloud droplet activation scheme that includes the SVOCs, it is the cloud base size distribution that is needed and the DCP can be used to calculate such a size distribution irrespective of the initial relative humidity. The same linear RH profile that was used to derive $t_{cb}$ can be used to derive approximate times to reach cloud base from any initial relative humidity, which we define as $t_{cb}^*$. A discussion of the best initial RH to use for the DCP is available in the supplementary material and is found to be 99.9% so that the cloud base time used in the DCP is

$$t_{cb}^* = \frac{100 - 99.9}{100 - RH_0} \times t_{cb}, \tag{B6}$$

where $RH_0 = S_0 \times 100\%$. This incorporates the fact that there is no significant difference in equilibrium condensed mass until the relative humidity approaches 100% compared to 90% RH as well as the fact that the condensation rates derived in the DCP are only approximate.

**Appendix C: Mass Loadings**

Stated here are the total mass concentrations used in the four different environmental conditions under the three different mass loadings. The mass is distributed between the volatility bins according to Table 3.

**Table 4.** Total mass concentrations of the semi-volatile organic compounds used in the four different environmental conditions. Values are in $\mu$g m$^{-3}$.

| Case | Natural | Rural | Near-City | Urban |
|------|---------|---------|-----------|----------|
| 10% | 3.2758 | 3.3814 | 4.3204 | 4.4955 |
| 50% | 27.0228 | 27.8838 | 36.3240 | 37.8741 |
| 90% | 212.4513 | 219.0913 | 288.1826 | 302.2525 |

*Acknowledgements.* The research leading to these results has received funding from the European Union's Seventh Framework Programme (FP7/2007-2013), under grant agreement n° 603445 and the NERC funded CCN-vol (NE/L007827/1) project.



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
