# Peer review of "A parameterisation for the co-condensation of semi-volatile organics into multiple aerosol particle modes"

_Geoscientific Model Development, 2017_

## Short Comment (SC1) · 18 Jul 2017

Dear authors,

in my role as Executive editor of GMD, I would like to bring to your attention our Editorial version 1.1:

http://www.geosci-model-dev.net/8/3487/2015/gmd-8-3487-2015.html

This highlights some requirements of papers published in GMD, which is also available on the GMD website in the 'Manuscript Types' section:

http://www.geoscientific-model-development.net/submission/manuscript_types.html

[Figure]

In particular, please note that for your paper, the following requirements have not been met in the Discussions paper:

- "The main paper must give the model name and version number (or other unique identifier) in the title."

- "If the model development relates to a single model then the model name and the version number must be included in the title of the paper. If the main intention of an article is to make a general (i.e. model independent) statement about the usefulness of a new development, but the usefulness is shown with the help of one specific model, the model name and version number must be stated in the title. The title could have a form such as, "Title outlining amazing generic advance: a case study with Model XXX (version Y)"."

In order to simplify reference to your developments, please add the parametrisation name (and/or its acronyms) and a respective version number in the title of your article in your revised submission to GMD.

Yours,

Astrid Kerkweg

———————————————————

---

## Referee Comment (RC1) · Anonymous Referee #1 · 18 Sep 2017

This work extends a model to account for the condensation of semi-volatile organic vapors (SVOCs) in cloud droplet activations parameterizations to multimodal aerosol populations. The authors use a dynamic approach to account for the condensation of volatile species onto preexisting aerosol particles, and parameterize their effect on droplet activation as an increase in the mean dry particle diameter at cloud base. The performance of the new model is evaluated using a well known CCN activation parameterization and comparing against parcel model simulations. In general condensation of SVOCs onto aerosol particles have a modest effect on droplet number, although in some polluted cases seems to be more important. The paper is written in considerable detail and addresses important points, relevant for the scientific community. I

recommend its publication in GMD after a few minor comments are addressed.

General Comments:

- Not only the size of the wet particles changes with the condensation of SVOCs but also their composition. This is completely missing in the manuscript but may be quite important. Modes changes in the surface tension of the wet particles may have a much larger impact on CCN activation than large changes in particle size.

- The approach that results in Equation (17) is still monodisperse in that it does not take into account the variation in the condensation rate due to variation in the droplet size distribution (e.q., the equation should be weighted by the size distribution of the wet aerosol). Please note this and estimate its impact on the resulting activated fraction.

Technical comments:

Line 25. It must be "the" precipitation rate.

Line 50. There is an even more recent version of the Fountokis and Nenes (2005) parameterization. See: Morales Betancourt, R. and Nenes, A.: Droplet activation parameterization: the population-splitting concept revisited, Geosci. Model Dev., 7, 2345-2357, https://doi.org/10.5194/gmd-7-2345-2014, 2014.

Line 120. Remove the words "that is".

Line 156. Is this wet or dry diameter?

Line 409 and Figure 6. DCP seems to overpredict the activated fraction at high aerosol concentration.

Figures 3-7. Please add a legend to the Figures to improve readability.

---

## Referee Comment (RC2) · Anonymous Referee #2 · 6 Nov 2017

**1   Recommendation**

The authors develop a new parameterisation for cloud droplet activation that takes into account condensation of SVOCs on the aerosol and works for multiple modes. The derivation of the parameterisation is explained with some detail (but can be improved, see Major points), and the new parameterisation is thoroughly tested by comparing its results with results from a parcel model. The paper should be published after a couple of improvements and corrections that can easily be achieved.

[Figure]

**2 Major points**

Section 2.1: What is the difference between "equilibrium partial pressure" and "saturation partial pressure"? Is it the Kelvin factor? Another similar expression is "equilibrium saturation ratio". This is puzzling. Please clarify.

Section 2.2:
1) It took me quite a while to understand the derivations in this section. First, equation 1 is a bit hard to understand unless one knows that $y_i$ is the condensed mass *of a single particle* per unit mass of air. It is also a bit puzzling that while everything else is "per unit mass of air", $N$ is simply the number of particles. Although checking the units in eqs. 1 and 4 confirms that $N$ is indeed a simple number, I feel uncertain what to choose for $N$ in a concrete case. At least in your examples you use it as per cubic centimetre. I think it would be more natural to let $N$ be the number of particles per unit mass of air (or unit volume) and $y$ simply the condensed mass of a particle.
2) What in these derivations do you mean exactly with "initial"? Is this a more or less randomly chosen initial state or something fixed? The equations show that results depend on the initial state. Thus it seems important to know how it is chosen.
3) It should be explained why the equilibrium partial pressure can be neglected in the derivation of eq. 5.

Section 2.3: I was quite confused by your use of the "dummy index". This is not necessary at all and the sentence after eq. 9 should be deleted. Eq. 9 would have the same meaning with the summation over $j$ instead of $k$. The confusion is increased by two typos in equations 9 and 10 where instead of $y_{ij}^0$ I think it should be $y_{ik}^0$ under the sum over index $k$.

**3   Minor points**

Page 1, line 9: check unit! I assume that instead of $\mu g^{-3}$ $\mu g\,m^{-3}$ is meant.

Page 6, lines 1,2: The sentence beginning with "At the high..." is unclear. What is the "it" in "... and reduces it..."? Please reformulate.

Page 6, line 156: better define $D_d$ as "dry diameter", to distinguish it from later uses of "diameter".

Eq. 20 and equations in A1 and A2: while these equations are correct their form is unnecessarily complicated. Why don't you just write

$$SD = D_m\,\sqrt{\sigma}\,\sqrt{(\ln\sigma - 1)}$$

in Eq. 20 instead of the complicated expressions involving exponential functions and logarithms? I assume that in your code you do not compute the exp of logs since that would take unnecessary CPU time.

Page 12, lines 326,327: I don't know what you mean here. Is RH your initial RH so that problems occur when your initial RH is too close to 100%? Otherwise, approaching cloud base you will always end up with RH near 100%.

Page 12, line 331: form → from.

Page 13, line 335: "is the same as the ammonium sulphate modes". Which ammonium sulphate modes? It seems here to me as if I had missed something before, but the ammonium sulphate appears here for the first time.

Page 18, line 433: delete "a" before "carbonaceous".

Page 18, line 441: I assume you mean hydrophobic here (last word in the line).

Page 20, line 475: as → at (90%...).

Page 28, line 696: delete the dot before the comma.

---

## Author Comment (AC1) · 8 Feb 2018

Author Responses to Reviewer 1

1. Indeed, the amount of and rate of condensation of both water and the SVOCs is influenced by the size of the particles and their composition. This effect, however, is highly non-linear and unlikely to ever be accurately represented in a computationally efficient parameterisation. Our approach is to derive a model for the dynamic condensation of SVOCs near cloud base that assumes a quasi-constant composition. The aerosol particle size distribution that is used in the activation scheme of Fountoukis and Nenes, however, does take into account both the resulting change in size and chemical composition that this condensation of SVOCs induces. We use a mass averaged approach (using the condensed masses from the DCP) to calculate the change in chemical composition. Therefore, we appreciate that our parameterisation is an approximation but we do believe that changes in chemical composition have been included in the CCN activation process to an effective degree.

With regards to changes in surface tension, this is an active area of research and not a factor that definitively should be included in such models. Again, our approach is an approximation and some details must always be neglected in order to derive computationally efficient models.

An ideal parameterisation would take into account, for example, the geometric standard deviation of a lognormally distributed size distribution of particles, however, we have chosen to use an "effective" diameter in our calculations. We have chosen to use the median diameter in this setting to calculate the total condensed mass of SVOCs on a particular particle mode. This mass is, however, distributed between different sizes of particles based on their geometric standard deviation. The method of doing this is given in Connolly et al. (2014) and assumes a constant arithmetic standard deviation but varies (reduces) the geometric standard deviation while conserving mass (involatile plus SVOCs). The result is a narrower size distribution at cloud base that is then inserted into the activation scheme of Fountoukis and Nenes. Consequently, we do believe that we have made use of the particular particle size distribution through changes in the geometric standard deviation.

Line 25: corrected Line 50: We really meant just those based on the Fountoukis and Nenes with its later derivatives using a similar approach. We have modified the sentence to account for this. Line 120: words removed Line 156: word "dry" added Line 409 and Figure 6: We disagree, the parcel model results are higher than the parameterisation in Figure 6 Figures 3-7. We feel that adding a legend would actually clutter the graphs especially as there is no concise way of describing some of the models. As there are multiple similar figures that all use the same colour of line for each model

we think that an initial investment of time to understand the first graph will allow all subsequent graphs to be understood.

Author Responses to Reviewer 2

Yes, the Kelvin term distinguishes the equilibrium from the saturation partial pressure.

1. N should be number concentration and has been updated. 2. By initial, we mean the state at which the cloud activation scheme of Fountoukis and Nenes is implemented. This will depend on the specific large-scale model used and as such we have kept it in the general setting. An explanation of this has been inserted at line 168. 3. The justification for neglecting the equilibrium partial pressure is explained in detail in Section 2.1. 4. Section 2.3: The use of the phrase "dummy index" means that the index k doesn't mean anything physically and is used only in the summation. Unlike, I that refers to a specific particle mode and j that refers to a specific volatility bin. In equation 9, the use of the index k in the summation is, indeed, redundant but in equation 10 there is both a particular volatility bin with index j and a summation over all volatility bins and we feel that the use of k is important here. There are, indeed, typos in equation 10 that have been corrected and hopefully resolve the issue.

Page 1, line 9: corrected Page 6, lines 1,2: The use of "high RH" is deliberately vague as this is an assumption made in the model and it is difficult to quantify for a general case. Instead, we make an assumption and compare the results on CCN concentration of making such an assumption. The use of "it" has been clarified. Page 6, line 156: word "dry" added Eq. 20: The geometric standard deviation of the aerosol size distribution at cloud base (including SVOCs) is different to that of the involatile particles. That is why there are multiple subscripts in order to distinguish them. The use of $\ln\sigma$ is notational and refers in its entirety to the geometric standard deviation. We do not, in fact, calculate exponentials of logs. Page 12, lines 326,327: We have added "in a dynamic model" to clarify this. But we mean the condensed mass doesn't change much between an RH of 90% and 91% but changes a lot between 99% and 100%. Page 12,

line 331: corrected Page 13, line 335: correct to "involatile particle mode" Page 18, line 433: corrected Page 18, line 441: corrected Page 20, line 475: "as" changed to "at" Page 28, line 696: corrected

––––––––––––––––––––––––––––––

---

## Author Response (AR2)

[revised manuscript text omitted]

Checked through the manuscript and made some corrections